# The development of synaptic transmission is time-locked to early social behaviors in rats

Shovan Naskar[1,2], Roberto Narducci [1], Edoardo Balzani[3,2], Andrzej W. Cwetsch [1,2], Valter Tucci[3] & Laura Cancedda [1,4]

The development of functional synapses is a sequential process preserved across many brain areas. Here, we show that glutamatergic postsynaptic currents anticipated GABAergic currents in Layer II/III of the rat neocortex, in contrast to the pattern described for other brain areas. The frequencies of both glutamatergic and GABAergic currents increased abruptly at the beginning of the second postnatal week, supported by a serotonin upsurge. Integrative behaviors arose on postnatal day (P)9, while most motor and sensory behaviors, which are fundamental for pup survival, were already in place at approximately P7. A reduction in serotonin reuptake accelerated the development of functional synapses and integrative huddling behavior, while sparing motor and sensory function development. A decrease in synaptic transmission in Layer II/III induced by a chemogenetic approach only inhibited huddling. Thus, precise developmental sequences mediate early, socially directed behaviors for which neurotransmission and its modulation in supragranular cortical layers play key roles.

[1] Local Micro-environment and Brain Development Laboratory, Istituto Italiano di Tecnologia, Genova, Italy. [2] Università degli Studi di Genova, Via Balbi, 5, 16126 Genova, Italy. [3] Genetics and Epigenetics of Behaviour Laboratory, Istituto Italiano di Tecnologia, Via Morego, 30, Genova 16163, Italy. [4] Dulbecco Telethon Institute, Via Varese 16b, Roma 00185, Italy. These authors contributed equally: Shovan Naskar, Roberto Narducci. These authors jointly supervised this work: Valter Tucci, Laura Cancedda. Correspondence and requests for materials should be addressed to V.T. (email: valter.tucci@iit.it) or to L.C. (email: laura.cancedda@iit.it)

I nvestigations in developmental neurophysiology performed over the last decade have provided a general model of how timely, sequential events occur at cellular and network levels in a similar manner across diverse brain areas[1]. For example, a developmental sequence for synaptogenesis was reported, with GABAergic synapses becoming functional before glutamatergic synapses. In particular, previous studies in the hippocampus revealed that GABAergic signaling is depolarizing and mildly excitatory during the first postnatal week of development, and the maturation of the glutamatergic system lags behind the GABAergic system[2]. This developmental sequence is common to many brain regions[2], and allows neurons to mature at early stages driven by the mild excitation provided by GABA, while avoiding the toxic effects of strong excitation driven by glutamate[2,3].

Together with the developmental sequences for GABA and glutamate transmission, a sequence for the emergence of diverse neuronal network-driven, early patterns of coordinated activity in a number of brain areas[1,4,5] has also been reported. For example, in the immature cortex, two sequential synapse-driven network activity patterns exist. Cortical early network oscillations (cENOs), which present a large amplitude and low frequency oscillatory calcium waves, occur between P0 and P7[6]. These oscillations are primarily driven by NMDA and AMPA receptors, but not GABA$_A$ receptors. At approximately the end of the first week of postnatal life, recurrent patterns of large synaptic activity known as 'giant depolarizing potentials' (GDPs) driven by GABA$_A$-mediated conductances appear[6].

The depolarizing actions of GABA and early synchronous activity during the first postnatal week are pivotal for the morphological and functional maturation of neurons and the establishment of their first connections[7,8]. Then, the initial connections mature in complex neuronal networks toward the end of the first postnatal week, and their finely tuned activity begins to encode both primitive yet complex behaviors (e.g., reflexes, sensory, and motor functions) and subsequent integrative behaviors (e.g., social and cognitive) in mammals[1]. Accordingly, the very early patterns of coordinated neuronal activity are silenced in subcortical brain structures that govern movement, immediately before the pups begin to exhibit locomotion[1]. Indeed, only by then neuronal networks have developed to a point that early developmental programs are no longer required. However, researchers have not yet determined whether a developmental sequence also exists that actively promotes a timely switch from early patterns of coordinated neuronal activity essential for network development to finely tuned neuronal activity essential to encode and support complex behaviors (i.e., first reflexes and primitive functions, and then highly integrative behaviors that are ethologically relevant within the environment).

Here, we describe a peculiar and unprecedented temporal profile of functional synaptogenesis in the rat neocortex that we show to be relevant to the development of an early form of group behavior, huddling between littermates. We show that glutamatergic conductances anticipate GABAergic ones in the supragranular layer of the rat somatosensory cortex. Both currents abruptly increase during the second postnatal week with a temporal profile that matches the developmental profile of huddling between littermates. Huddling behavior depends on the activity of the somatosensory cortex, and both functional synaptogenesis and huddling are shifted toward earlier postnatal days by increasing brain serotonin levels. Our findings provide evidence for the association between region-specific timely neurodevelopmental processes and the emergence of complex behaviors relevant for sociability.

## Results

**Supragranular layers show abrupt synaptogenesis development.** We began our investigation by profiling the time course of synaptogenesis in the neocortex, the brain structure that controls the highest cognitive functions. We recorded spontaneous postsynaptic currents (sPSCs) from visually identified Layer II/III pyramidal neurons in acute brain slices of the somatosensory cortex of rat pups from P2 to P15. The frequency of spontaneous glutamatergic currents was significantly greater than 0 Hz at P5 ($0.54 \pm 0.19$ Hz). Conversely, the frequency of spontaneous GABAergic currents only was significantly different from 0 only later, at P7 ($0.20 \pm 0.11$ Hz; Fig. 1a). Notably, the frequencies of both glutamatergic and GABAergic spontaneous currents rapidly increased between P7 and P9 (glutamatergic currents: $0.88 \pm 0.18$ Hz at P7 vs $2.62 \pm 0.36$ Hz at P9; GABAergic currents: $0.20 \pm 0.11$ Hz at P7 vs $1.60 \pm 0.39$ Hz at P9; Fig. 1a). Next, to investigate the temporal sequence of glutamatergic and GABAergic synaptogenesis per se—separate from the level of network activity—we recorded pharmacologically isolated miniature postsynaptic currents (mPSCs). The frequency of both glutamatergic and GABAergic mPSCs from pyramidal neurons in supragranular layers of the neocortex displayed an abrupt increase between P8 and P9 (glutamatergic currents: $0.64 \pm 0.14$ Hz at P8 vs $2.83 \pm 0.36$ Hz at P9; GABAergic currents: $0.07 \pm 0.04$ Hz at P8 vs $1.49 \pm 0.48$ Hz at P9; Fig. 1b). These data are consistent with the increased number of cells characterized by low activity ('low activity cells' in Supplementary Fig. 1a, see the methods for a quantitative definition) when we recorded GABAergic conductance compared to glutamatergic conductance between P2 and P10 (Supplementary Fig. 1a). Then, we calculated the ratio between the normalized frequency of glutamate and GABA mPSCs (glutamate–GABA ratio; values extracted from the mPSC dataset, excluding the low activity cells) to more precisely identify the timing of the prevalence of glutamatergic and GABAergic conductance during the first week of postnatal life. The normalized glutamate–GABA frequency ratio remained greater than '1' in the superficial layers of the neocortex from P4 to P11 (Supplementary Fig. 1b), emphasizing the fact that the glutamatergic conductance appeared earlier than the GABAergic conductance.

**Diverse brain regions show specific synaptogenesis sequences.** Next, we investigated whether functional synaptogenesis also occurred in a glutamate–GABA sequence in the neocortical infragranular layers. Interestingly, glutamatergic synaptogenesis generally occurred mostly concurrently with GABAergic synaptogenesis and in a linear fashion in Layer V pyramidal neurons (Supplementary Fig. 1c). The analysis of the glutamate–GABA ratio revealed a more uniform distribution in the first postnatal week, with positive values only observed at P4 and P5 (Supplementary Fig. 1d).

The distinguishing aspect of how glutamatergic and GABAergic synapses develop over time and across different cortical layers was confirmed by recording GABAergic and glutamatergic mPSCs in the hippocampus, where the 'GABA-first/glutamate-after' sequence of synaptogenesis has been thoroughly characterized[9]. As expected, we found that GABAergic conductance preceded glutamatergic conductance in the neonatal hippocampus (Supplementary Fig. 1e and f).

Next, we investigated whether the upsurge in functional synaptogenesis in Layer II/III at the beginning of the second postnatal week had anatomical correlates. We counted spines on secondary dendrites of neurons that were pressure-injected with a lipophilic dye (DiI) between P7 and P15. Consistent with the electrophysiological recordings, the spine density in Layer II/III

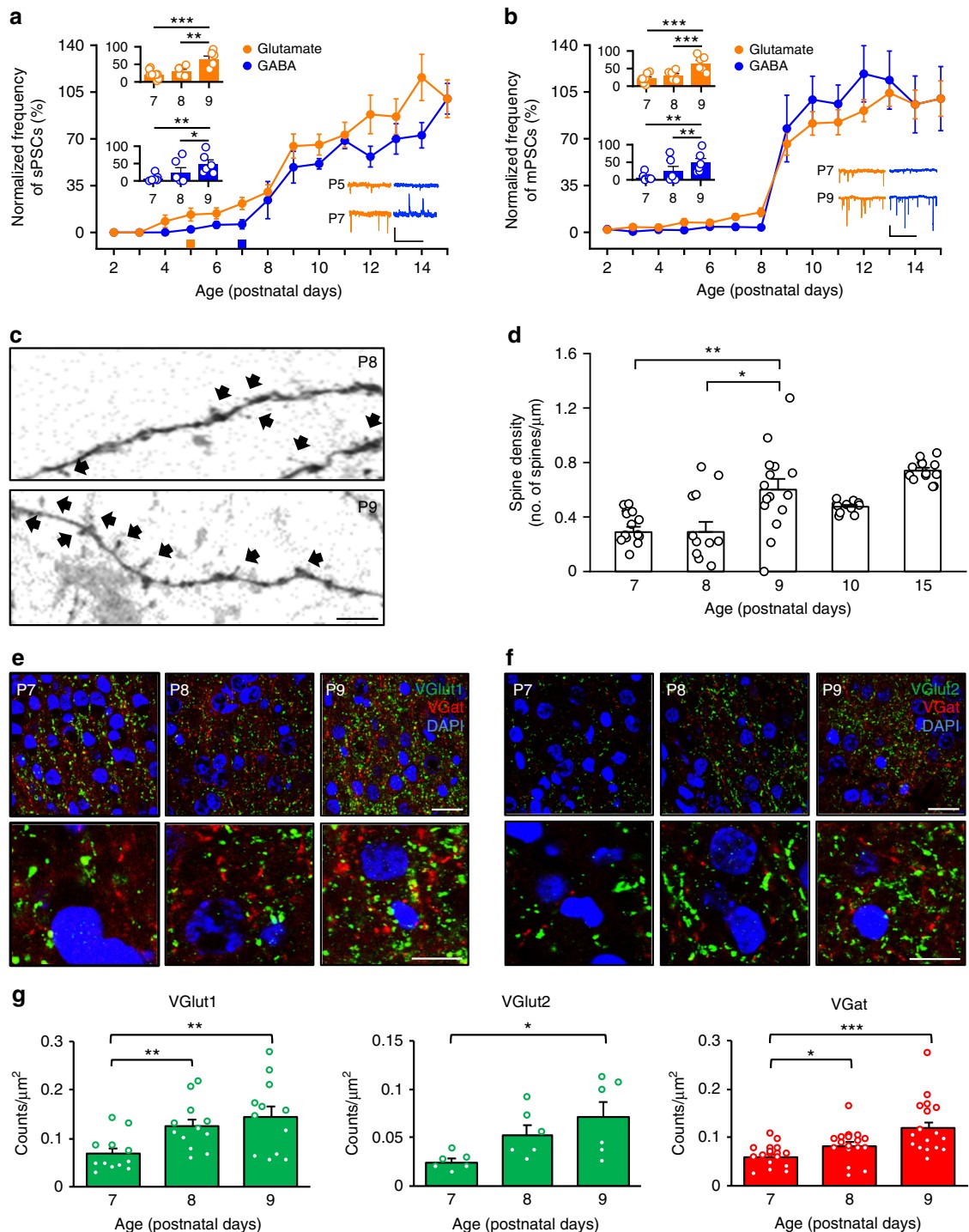

was significantly increased at P9 compared to P7 and P8 (0.29 ± 0.04 spines/μm at P7 and 0.29 ± 0.07 spines/μm at P8 vs 0.60 ± 0.08 spines/μm at P9; Fig. 1c, d), whereas we did not observe significant differences between P9, P10 and P15 (0.60 ± 0.08 spines/μm at P9, 0.48 ± 0.01 spines/μm at P10, 0.74 ± 0.02 spines/μm at P15, Fig. 1d). These findings were confirmed by performing a Golgi–Cox staining-based spine counting method, where we also observed a significantly increased spine density between P7 and P9 (0.43 ± 0.04 spines/μm at P7 vs 0.84 ± 0.03 spines/μm at P9; Supplementary Fig. 2a and b). Conversely, when we performed the same DiI experiments in Layer V, we found a progressive increase between P8 and P15, consistent with our electrophysiological data (Supplementary Fig. 2c and d).

Next, we also counted the number of puncta that were stained positive for presynaptic glutamate (VGlut-1 and VGlut-2) and GABA (VGat) transporters in the supragranular layers of the neocortex and found that the numbers of puncta expressing each of these proteins (presented as counts/μm²) increased significantly between P7 and P9, consistent with our electrophysiological and anatomical observations (VGlut-1: 0.06 ± 0.01 at P7 vs 0.14 ± 0.02 at P9; VGlut-2: 0.02 ± 0.003 at P7 vs 0.07 ± 0.01 at P9; VGat: 0.05 ± 0.005 at P7 vs 0.11 ± 0.01 at P9; Fig. 1e–g).

**Motor/sensory functions anticipate cortical synaptogenesis.** From a behavioral perspective, the first 2 weeks of postnatal life in rodents are crucial for the establishment of behaviors that will

**Fig. 1** Synaptic currents in neocortical Layer II/III sharply increase after P7, which correlates with changes in spine density and presynaptic markers. **a** Development of glutamatergic (orange) and GABAergic (blue) sPSCs ± SEM; $N = 83$ layer II/III neurons from 26 animals (2–11 neurons per postnatal age). Values were normalized to the mean sPSCs recorded at P15. Squares at the bottom indicate sPSC frequencies significantly greater than 0 Hz (one-tailed one-sample Wilcoxon-signed rank test against a theoretical value of 0, $W = 15$, $P < 0.05$). Left inset: Single data points and average glutamatergic and GABAergic sPSCs ± SEM recorded between P7 and P9. Glutamatergic sPSCs: one-way ANOVA, $F_{(2,17)} = 12.8$, $P < 0.001$, post hoc pairwise $t$-test comparisons with Holm's correction. GABAergic sPSCs: Kruskal–Wallis test $\chi^2_{(2)} = 9.3$, $P < 0.05$, post hoc Dunn's test with Holm's correction. Right inset: Representative traces of glutamatergic and GABAergic sPSCs. Scale bars: 20 pA, 2 s. **b** Development of glutamatergic and GABAergic mPSCs ± SEM; $N = 158$ neurons from 54 animals (9–14 neurons per postnatal age). Values were normalized to the mean mPSCs recorded at P15. Left inset: same as in **a** but for glutamatergic and GABAergic mPSCs. Kruskal–Wallis test, $\chi^2_{(2)} = 21$, $P < 0.001$ and $\chi^2_{(2)} = 11$, $P < 0.01$, respectively, post hoc Dunn's test with Holm's correction. Right inset: Representative traces of glutamatergic and GABAergic mPSCs. Scale bars: 20 pA, 2 s. **c** Confocal images of secondary dendrites of Layer II/III neurons stained with DiI. Black arrows indicate spine-like protrusions. Scale bar: 5 μm. **d** Average spine density ± SEM and single data points for dendritic branches (10 animals total; Kruskal–Wallis test, $\chi^2_{(4)} = 34$, $P < 0.001$, post hoc Dunn's test against P9 with Holm's correction). **e**, **f** Confocal images (top panels) and high-magnification images (bottom panels) of somatosensory cortical slices stained for VGlut-1, VGlut-2 (green), VGat (red), and DAPI (blue). Scale bars: 50 μm, top panels; 25 μm, bottom panels. **g** Average count density ± SEM and single slice data points for experiments as in **e** and **f** (18 animals total; Kruskal–Wallis test, $\chi^2_{(2)} = 11$, $P < 0.01$, $\chi^2_{(2)} = 6.9$, $P < 0.05$, and $\chi^2_{(2)} = 16$, $P < 0.001$, respectively, post hoc Dunn's test with Holm's correction). *$P < 0.05$, **$P < 0.01$, and ***$P < 0.001$

initially guarantee the survival of the pup within the environment and subsequently promote interactions within peers[10]. Therefore, we next investigated the development of motor and sensory behaviors in neonatal rats to identify a functional behavioral correlate of the timeline of synaptic development. To this aim, we adapted the SHIRPA (SmithKline Beecham, Harwell, Imperial College and Royal London Hospital Phenotype Assessment) neurological test battery (originally developed as a generalized neurological tool for the assessment of phenotypes in neurologically challenged adult rodents) to rat pups (P2–P10)[11,12]. We analyzed 21 different behaviors (Supplementary Table 1) and then performed a principal component analysis (PCA) with a k-means of '2' (see the methods). The analysis identified two different behavioral clusters that we defined as: primitive reflexes (fully developed already at P2), and motor/sensory functions (mostly developing between P6 and P8). Then, we further split the motor/sensory classification in two separate classes (i.e., motor and sensory) based on the phenotypic domain (Supplementary Table 1).

The development of primitive reflexes, motor and sensory functions preceded cortical functional synaptogenesis, consistent with the fact that most of the behaviors screened with the modified SHIRPA test were reflexive behaviors and possibly involved subcortical neural structures rather than the cortex (Fig. 2a, b and Supplementary Fig. 3a).

**Huddling development matches cortical synaptogenesis**. We next searched for more ethologically meaningful behaviors, which involve the integration of cortical circuits. Huddling is a spontaneous, interactive, natural group-dependent behavior that manifests in many species, including infant rats. This behavior is associated with the formation and maintenance of clumps among littermates. We adapted a previously described strategy to trigger huddling in pups under controlled experimental conditions[13]. We separated rat pups from their dams on consecutive days (P2–10) and video-recorded them for 10 min while housing the litter in an empty arena (Fig. 2c). The offline analysis of huddling among littermates allowed us to derive three different measures (time spent together, no. of different clusters, and no. of cluster switches) to quantify this behavior (Supplementary movie 1 and 2; Supplementary Table 2). The occurrence of all 3 analyzed huddling measures sharply increased between P8 and P9 (Fig. 2c, d and Supplementary Fig. 4a; left panel: 0.76 ± 0.18 min. at P8 vs 2.41 ± 0.51 min. at P9; middle panel: 4.12 ± 0.58 at P8 vs 13.75 ± 2.1 at P9; and right panel: 0.25 ± 0.16 at P8 vs 4.75 ± 1.19 at P9), which correlated with the sharp increase in mPSCs observed between P8 and P9 (Fig. 1b).

**Huddling is driven by interaction with peers**. Most postnatal behaviors depend on the development of the motor system. Moreover, huddling behavior is traditionally associated with a thermoregulatory need of altricial pups[14]. We thus paralleled our huddling study with an assessment of both motor and temperature measures to test the hypotheses that the development of huddling was either simply an outcome of the motor control or the need to remain warm in the absence of the mother. Should any of the two (i.e., motor and thermoregulatory) hypotheses fail, the remaining interpretation was a social, integrative behavior driven by the need to actively interact with peers. When we examined the space the pups occupied within the huddling arena as a measure of the overall movement of pups during a putative experimental session, we observed the highest occupation at P7, with decreased movement afterwards (0.68 ± 0.02% at P7 vs 0.48 ± 0.01% at P9; Supplementary Fig. 5a; Supplementary Table 2). Thus, the developmental profile of the motor skills differed from the developmental profile of huddling behavior, which was fully developed only by P9. Moreover, huddling behavior was not driven by a decrease in body temperature upon separation from the dam. Indeed, when we measured the temperature change between the beginning and the end of the experimental sessions in non-huddling animals across ages, we observed an age-dependent decrease (Pearson's $r = 0.68$, $P < 0.001$; Supplementary Fig. 5b). The observation that the pups that were separated from the mother showed a lower decrease in temperature with increasing ages exhibits an opposite trend to the developmental profile of huddling behavior, which increased with age.

**Somatosensory cortex activity is required for huddling**. Next, we investigated whether cortical activity was indeed necessary for huddling behavior. Using in utero electroporation at embryonic day (E) 17.5, we transfected pups with control constructs (GFP only) or an inhibitory DREADD (hM4Di, iDREADD) in a subpopulation of precursor cells committed to become pyramidal neurons of Layer II/III in the somatosensory cortex[15,16] (Fig. 3a–c). We performed electrophysiological recordings before and after a bath perfusion of the DREADD activator clozapine-N-oxide (CNO, 10 μM) in acute brain slices collected at P9 from pups that had been electroporated with control constructs and iDREADDs to verify the ability of iDREADDs to exert its inhibitory effect on cortical activity. As predicted in the literature[17], the CNO application significantly reduced the number of action potentials in iDREADD-expressing neurons (2.7 ± 0.6 preCNO vs 0.43 ± 0.2 postCNO), with no significant effect on control neurons (1.89 ± 0.45 preCNO vs 1.89 ± 0.59 postCNO; Fig. 3d). After CNO administration, iDREADD-expressing neurons also

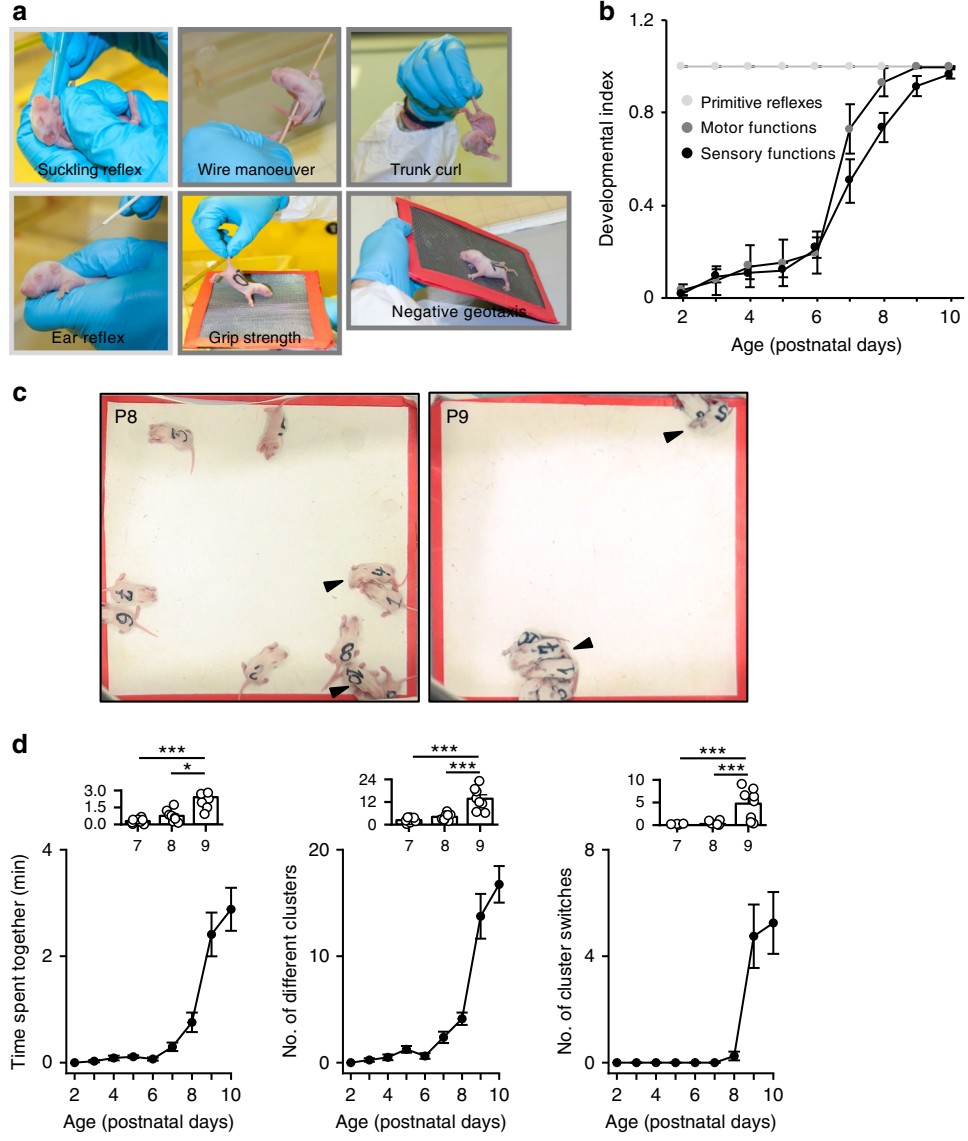

**Fig. 2** Huddling behavior develops at the beginning of the second postnatal week, whereas primitive reflexes and motor and sensory functions develop earlier. **a** Representative images showing the primitive reflexes and motor and sensory functions, which are color-coded as described in **b**. **b** Developmental indexes for primitive reflexes (light gray) and motor (dark gray) and sensory functions (black) ($N = 80$ animals from 8 litters). **c** Representative images of huddling behavior of animals of different ages when experimental sessions ended. Black arrowheads indicate individual clusters. **d** Quantification of the average litter values ± SEM for the three parameters (time spent together, no. of different clusters, and no. of cluster switches), see the methods for the quantification of huddling behavior. Left panel: Kruskal–Wallis test, $\chi^2_{(2)} = 16.3$, $P < 0.001$, post hoc Dunn's test with Holm's correction; middle panel: one-way ANOVA, $F_{(1,22)} = 22.8$, $P < 0.001$, pairwise $t$-test comparisons with Holm's correction; right panel: type II ANOVA of ranks, $F_{(1,22)} = 13.2$, $P < 0.001$, h3-corrected. Top inset: average values and single litter data point for P7–9 of the data below. *$P < 0.05$ and ***$P < 0.001$. Tests shown in **b** and **d** were performed on the same set of animals

exhibited a significant reduction in the mPSCs frequency (2.73 ± 0.23 Hz preCNO vs 1.11 ± 0.39 Hz postCNO), whereas control vector-expressing neurons maintained their synaptic activity unaltered (1.98 ± 0.33 Hz preCNO vs 1.99 ± 0.29 Hz postCNO, Fig. 3e).

Thus, we assessed huddling before and after CNO treatment in litters where we electroporated all pups with iDREADDs to specifically address huddling in the presence of cortical inhibition. As controls, we utilized litters where all pups were electroporated with control vectors (GFP only). Litters transfected with iDREADDs showed impaired huddling behaviors (time spent together: 3.95 ± 0.54 min preCNO vs 2.60 ± 0.43 min postCNO; no. of different clusters: 10.00 ± 1.21 preCNO vs 6.00 ± 0.70 postCNO; no. of cluster switches: 2.36 ± 0.54 preCNO vs 0.27 ±

0.27 postCNO; Fig. 3i and Supplementary Figs. 4c and 6b). Control litters did not show any significant changes (time spent together: 3.88 ± 0.64 min preCNO vs 3.94 ± 0.38 min postCNO; no. of different clusters: 8.78 ± 0.88 preCNO vs 8.67 ± 0.74 postCNO; no. of cluster switches: 1.33 ± 0.47 preCNO vs 1.78 ± 0.43 postCNO; Fig. 3g and Supplementary Fig. 6b). No alterations in the developmental index were observed in any group (Supplementary Table 4; Fig. 3f, h, Supplementary Figs. 3c, 6a).

Next, we investigated huddling before and after CNO treatment in litters where half of the embryos was electroporated with iDREADDs and the other half was electroporated with control vector (control and iDREADDs mixed litters). Also in control and iDREADDs mixed litters, we observed a statistically significant decrease in all the three huddling parameters 30 min

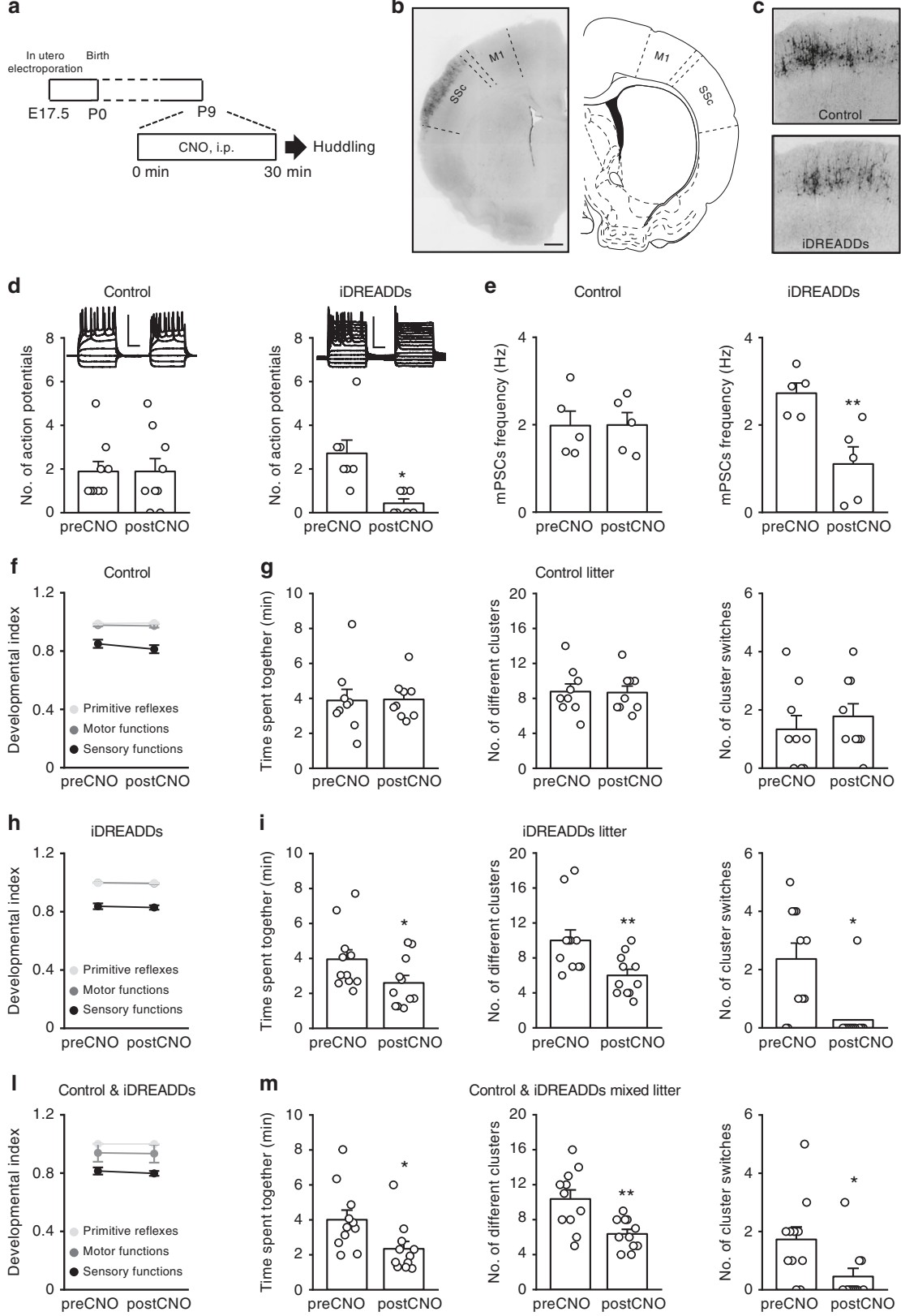

after the CNO injection (10 mg/kg, i.p.; P9; time spent together: $4.00 \pm 0.55$ min preCNO vs $2.34 \pm 0.42$ min postCNO; no. of different clusters: $10.36 \pm 1.04$ preCNO vs $6.36 \pm 0.53$ postCNO; no. of cluster switches: $1.72 \pm 0.43$ preCNO vs $0.45 \pm 0.28$ postCNO; Fig. 3m and Supplementary Fig. 6b), with no alterations in the developmental index (Supplementary Table 4;

Fig. 3l, Supplementary Figs. 3c, 6a). The observation that control pups from control and iDREADDs mixed litters (like iDREADDs-only litters) exhibited a reduction in huddling parameters is consistent with the fact that huddling is a group behavior. Indeed, being part of a litter where some of the littermates have impaired huddling behavior (the iDREADDs-

**Fig. 3** Huddling behavior depends on the activity of the somatosensory cortex. **a** Schematic depicting the experimental protocol. **b** Representative images of GFP fluorescence showing the expression of iDREADDs restricted to neurons of the somatosensory cortex (SSc) in in utero electroporation experiment (M1 is for primary motor cortex); image adapted with permission from the 'Brain Maps Atlas 4.0'[49]. Scale bar: 400 μm. **c** Higher magnification of images as in **b**. Scale bar: 40 μm. **d** Single neuron data points and average mPSC frequency ± SEM for layer II/III neurons electroporated with iDREADD before and after CNO application ($N = 7$ for iDREADDs plus GFP and $N = 9$ for GFP-only controls; 4 animals). **e** Single neuron data points and average number of action potentials ± SEM ($N = 5$ for iDREADDs plus GFP and $N = 5$ for GFP-only controls, 6 animals), upon CNO administration. **f**, **h**, **l** Mean litter developmental index ± SEM for primitive reflexes (light gray) and motor (dark gray) and sensory functions (black) before and after CNO administration at P9 in litters electroporated in utero with control vectors ($N = 9$ litters, 90 animals total), iDREADDs ($N = 11$ litters, 110 animals total) and control and iDREADDs mixed litters with half of the pups electroporated with control vector and half of the pups electroporated with iDREADDs ($N = 11$ litters, 110 animals total). see also Supplementary Table 4 for descriptive values, Supplementary Fig. 3c for relative polar plots and Supplementary Fig. 6a. **g**, **i**, **m** Parameters describing the huddling behavior at P9 before and after CNO administration in the same sets of animals as examined in **f**, **h**, and **l**. See Supplementary Table 3 for a complete list of statistical test, test statistic and degrees of freedom for Fig. 3; see also Supplementary Fig. 6b. *$P < 0.05$ and **$P < 0.01$

transfected pups) will affect the huddling behavior also of control pups during their common huddling session. To specifically assess this issue, we performed a single-pup analysis of all litters in which there were present both control and iDREADDs-transfected littermates by considering only individual pup huddling parameters ('time spent together' and 'no. of cluster switches'; see Supplementary Table 2) and classifying all pups in either the iDREADDs or control group. After the CNO injection, time spent together and the number of cluster switches significantly decreased in both iDREADD and control pups (time spent together: control pups: 4.47 ± 0.34 min preCNO vs 2.23 ± 0.20 min postCNO; iDREADDs pups: 3.97 ± 0.27 min preCNO vs 2.65 ± 0.20 min postCNO; Supplementary Fig. 6c). Number of cluster switches: control pups: 1.00 ± 0.05 preCNO vs 0.19 ± 0.03 postCNO; iDREADDs pups: 1.14 ± 0.06 preCNO vs 0.24 ± 0.03 postCNO; Supplementary Fig. 6d. Altogether, these results indicate that the development of proper neuronal activity within cortical circuits is fundamental for the expression of huddling, which is a group behavior.

**Serotoninergic transmission affects synaptogenesis and huddling onset.** Serotoninergic (5-HT) neurotransmission has a central role in cortical development with both excess[18,19] and deficient[20,21] transmission exerting detrimental effects on physiological developmental programs. In particular, 5-HT axons densely innervate Layers I–IV of sensory cortices (and not the motor cortex) during a well-defined and transient phase when exuberant synaptogenesis is occurring (from P2.5 to P13–21 in rats[22,23]). Thus, we explored regulatory neurotransmission in the supragranular layers of the somatosensory cortex to obtain mechanistic insights that may account for the tight temporal modulations of both cortical synaptogenesis and integrative behaviors. We performed immunohistochemical staining for the serotonin transporter (SERT)[24], which is present on thalamocortical axons that are able to reuptake serotonin in the somatosensory cortex during development[25]. As predicted, we observed a significant increase in the innervation of the supragranular layers by SERT-positive (SERT+) processes between P7–9 (Supplementary Fig. 7). Thus, we interfered with 5-HT uptake early in development by systemically administering a selective serotonin reuptake inhibitor (SSRI), citalopram (CTP) to investigate whether a 5-HT 'upsurge' during the critical time window of P7–9 has a role in regulating synaptogenesis in the supragranular layers of the somatosensory cortex. We injected neonatal rats from P2 to P15 with CTP (10 mg/kg, i.p.[26–28]) and recorded mPSCs from neurons in Layer II/III of the somatosensory cortex at various time points. When we plotted excitatory and inhibitory mPSCs as a cumulative curve of the average frequency recorded on each postnatal day, we observed a shift in the curve of CTP-treated animals toward younger ages compared to controls (Fig. 4a, b). More precisely, we

detected the largest divergence between CTP and control groups at P8. This change was evidenced by the bar plots showing the difference between the CTP and control cumulative curves for each postnatal day (Fig. 4c, d). Finally, we performed the modified SHIRPA and huddling tests in CTP-treated animals from P2 to P10 to investigate whether 5-HT neurotransmission also influenced the onset of complex behaviors. Consistent with our electrophysiological data, the cumulative profiles of motor and sensory behaviors were similar between CTP-treated animals and age-matched controls (Fig. 4e, f and Supplementary Fig. 3a and b). Moreover, the cumulative curves of all three parameters describing the huddling behavior exhibited a leftward shift in CTP-treated animals, and reached the largest differences compared to controls at P8 (time spent together: 0.09 ± 0.04 min in controls vs 0.54 ± 0.14 min in CTP-treated animals at P4; no. of different clusters: 0.5 ± 0.27 in controls vs 5.4 ± 1.57 in CTP-treated animals at P4; no. of cluster switches: 0 in controls vs 1.4 ± 0.68 in CTP-treated animals at P6; Fig. 4g and Supplementary Fig. 4b). Importantly, the developmental trajectory of the space CTP-treated pups occupied within the huddling arena did not change compared to controls (statistical interaction between the factors 'treatment' and 'postnatal age' after a two-way ANOVA of ranks was not significant, $F_{(7,1023)} = 1.07$, $P > 0.05$; Supplementary Fig. 5c). Notably, the decrease in the body temperature of non-huddling pups observed after the CTP treatment retained the negative correlation with postnatal age (Pearson's $r = 0.55$, $P < 0.001$; Supplementary Fig. 5d).

## Discussion

Sensory information is driven from the periphery (e.g., skin or whiskers for somatosensory sensations) and transmitted to the thalamus, from which the bulk of thalamocortical fibers project into Layer IV of the primary sensory cortices. Layer IV neurons then project to Layer II/III, where sensory integration is achieved and signals are processed and transmitted to other brain regions through Layer V neurons or to Layer II/III of the contralateral cortex[29–32]. Although many studies have addressed the sequence of events that leads to the development of functional thalamocortical projections and the synapses they form, information on how functional synaptogenesis occurs in the cortex is much more scarce[33]. Moreover, the few reports that exist are not focused on the developmental time course of synaptogenesis, but rather on the relative sequence of development of GABAergic vs glutamatergic synapses and provided contrasting results. In particular, in the cortex, very few studies have directly confirmed the GABA-first/glutamate-after sequence reported for the other brain regions[34,35], whereas a number of other indirect pieces of evidence in fact suggested an earlier development of glutamatergic synapses[1,6,36–38]. Our study reveals a layer-specific (supragranular vs infragranular layer) sequence of glutamatergic and GABAergic

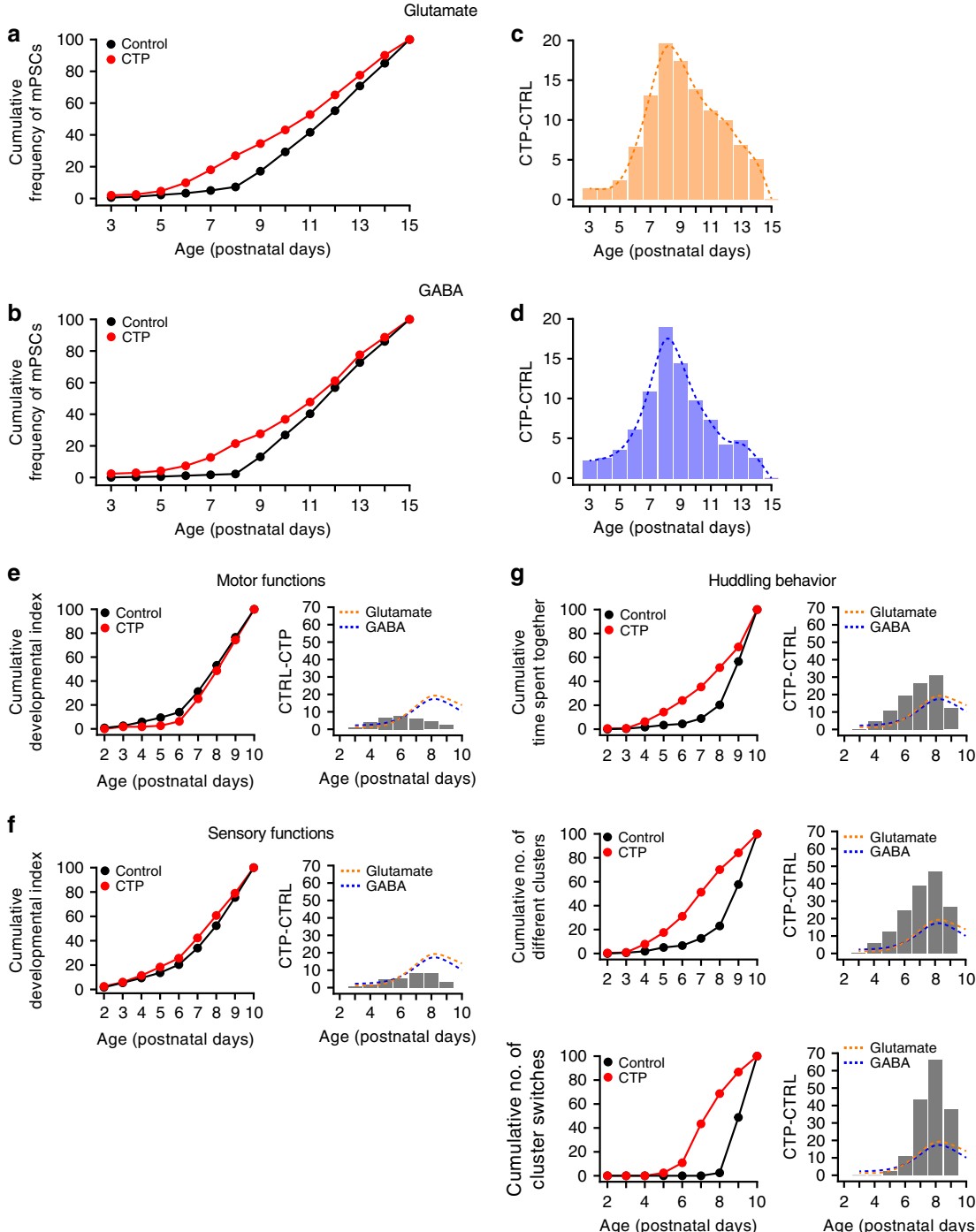

**Fig. 4** Blockade of 5-HT reuptake at early postnatal ages in vivo anticipates the temporal development of synaptic currents and the onset of huddling. **a**, **b** Cumulative developmental curves of the average frequencies of glutamatergic (**a**) and GABAergic (**b**) mPSCs from animals that were treated systemically with CTP (red, $N = 86$ cells from 13 animals) vs controls black traces (from Fig. 1b). **c**, **d** Bar plot showing the difference between the CTP and control cumulative curves for each postnatal day; dashed lines represent the polynomial fit of the difference distribution of glutamatergic (orange) and GABAergic (blue) mPSCs. The maximal separation between the control and the CTP cumulative curves was observed at P8 both for glutamatergic and GABAergic mPSCs. **e**, **f** Left panels: Cumulative developmental curves of motor and sensory functions. The CTP treatment did not alter the time course of the onset of motor and sensory functions (statistical interaction between the factors 'treatment' and 'postnatal age' after a two-way ANOVA of ranks was not significant for motor ($F_{(8,108)} = 0.29$, $P > 0.05$) and sensory functions ($F_{(8,198)} = 0.87$, $P > 0.05$); $N = 5$ litters, 50 animals total for CTP, and data for control from Fig. 2b). In (**e**), values from control were subtracted from CTP values to obtain positive differences. The cumulative curve for controls was calculated from the data presented in Fig. 2b. Right panel: bar plot showing the difference between the CTP and control cumulative curves for each postnatal day (gray bars). Bar plots were overlapped to dashed lines representing the differences observed for glutamatergic and GABAergic mPSC curves calculated in **c** and **d** for direct comparison. **g** Same as **e** and **f**, but for the three parameters describing the huddling behavior. For all parameters, the maximal difference between cumulative curves was observed at P8. Dotted lines represented the curves in **c** and **d** reported for direct comparison

synapse formation in cortical pyramidal neurons. Moreover, we found that the temporal development of synaptogenesis and spinogenesis occurs in a different manner among diverse cortical layers, with a linear trend in deep layers and abrupt development in the superficial layers.

What does this 'abrupt' increase in functional synapses relate to in physiological terms? Based on our electrophysiological and behavioral data, we speculate a model of cortical development where synaptogenesis is organized in a 'ready, steady, go' manner that may be central to the development of ethologically relevant behaviors in rodent pups. These behaviors would first be necessary for simple fostering and survival, then for experience-dependent development and eventually for social interactions with peers (Supplementary Fig. 8). Indeed, in the first postnatal week of life (by P6), both excitatory and inhibitory neurons have completed their migration from deeper to superficial layers and have populated the somatosensory cortex. Thalamocortical afferents carrying the bulk of sensory information from the thalamus are also in place by this time point[39]. We speculate that the somatosensory cortex is now possibly in a 'ready' state that would be sufficient for survival strategies (e.g., feeding and interaction with the dam), which require reflex responses but little movement. Consistent with this hypothesis, we observed that primitive reflexes that are possibly generated by subcortical structures were already present at birth. By the beginning of the second postnatal week (after P7), thalamocortical afferents start innervating Layers V and VI, enabling somatosensory perception from the environment[39]. Our results indicate that both glutamatergic and GABAergic currents increase steadily at this stage in these deeper layers, allowing the first processing of sensory information, and the cortex possibly enters a 'steady' state. We confirmed that the progression to the second week of postnatal life occurs concomitantly with the development of active motor and sensory functions, and pups start to explore the environment. This change allows experience-dependent development[40], but it also exposes the pup to the danger of predation when in the wild. Thus, at this stage, pups need to learn very quickly how to integrate motor and sensory patterns, to actively overcome possible dangers and to generate group behaviors to maximize social resources. Nevertheless, in the steady state, upper and more integrative layers (II and III), were still very quiet in the present study. Thus, it seems convenient that at P9—right after motor and sensory functions are completely developed—an abrupt increase in synaptogenesis occurs. We speculated that this would rapidly place the cortex in a safer 'go' state, enabling the pups to actively and promptly overcome the dangers from the environment, while making the best of positive interactions with peers. Indeed, only by P9 do integrative and social behaviors (like the novel huddling behavior that we describe here in pups) rapidly emerge.

The concomitant emergence of a 5-HT innervation in upper cortical layers[22,23] functions to accelerate the attainment of synaptic transmission in the integrative layers. Our hypothesis is supported by our own immunostaining results showing an increase in the number of 5-HT fibers at the beginning of the second postnatal week and by the earlier onset of synaptogenesis and huddling behavior in CTP-treated animals. Nevertheless, whether CTP (administered by i.p.) affects huddling behavior via a mechanism that is independent of the synaptic changes that we identified, remains an open question.

The establishment of neurotransmission in Layer II/III of the somatosensory cortex is an important factor mediating huddling behavior, which is independent of primitive reflexes, motor and sensory functions, and thermoregulation. This is demonstrated by our experiments showing an impairment in huddling after the administration of CNO to animals that were specifically electroporated with iDREADDs in the somatosensory cortex.

Interestingly, a neuronal subpopulation of the superficial somatosensory cortex receives and sends excitatory inputs to the limbic allocortex[41] (a region responsible for presenting a neural representation of social behaviors at the level of the neocortex[42]). Thus, we hypothesize that our electroporation strategy, which ensures the transfection of Layer II/III projection neurons of the somatosensory cortex, most likely targeted also this cohort of neurons, possibly affecting in turn huddling behavior. Finally, our study adds to the growing body of evidence that the primary somatosensory cortex, in addition to processing somatosensory information, also serves to integrate inputs from different sensory modalities (e.g., olfactory, visual, and motor functions[43–45]) with the possible function of initiating the organization of complex behaviors.

Huddling behavior has been observed in 67 different mammalian species, 40 of which are rodents[46]. Thus, the findings of the present study can be reasonably generalized to at least these models, assuming that a similar developmental timeframe of synaptogenesis affects altricial, social behaviors. Our study points out that the overall maturation of neuronal circuits is a temporally controlled process that exerts a profound influence over early integrative behaviors. For example, GABAergic synaptic maturation shapes the sensory integration of the mouse insular cortex (IC), a region involved in emotional and cognitive functions. In particular, in a mouse model manifesting social deficits, inhibitory neurons showed impaired postnatal maturation and weakened synaptic transmission. IC sensory integration was restored after neuronal $GABA_A$ receptor activity was increased through the injection of a positive allosteric modulator, but only during an early time window corresponding to circuit development[47]. This finding highlights the possible concomitancy between correct synaptic maturation and the emergence of social behaviors.

In conclusion, we showed that synaptogenesis in the somatosensory cortex is time-locked with huddling, which is the precursor of the complex social behavioral infrastructure of an individual. Thus, the results of our study implicate that timing of specific neurodevelopmental processes at the cellular level is pivotal for the emergence of complex behaviors. In the larger context of the human behavioral repertoire, our study predicts that a disruption of timing during the development of glutamatergic/GABAergic synapses may have dramatic consequences on sociability later in life.

## Methods

**Animal licenses and treatments**. All animal procedures were approved by IIT licensing, the Italian Ministry of Health (D.Lgs 26/2014) and EU guidelines (Directive 2010/63/EU). Sprague Dawley (SD) rats were housed in a room with a temperature of 21–22 °C, 12:12 light/dark cycle, food and water available ad libitum. For behavioral experiments involving iDREADDs or their GFP controls, all pups of the litter were injected at P9 with vehicle (saline solution, i.p., preCNO) and with clozapine N-Oxide (CNO, SIGMA, i.p., 10 mg/kg, postCNO), 30 min before subjecting them to each of the two huddling sessions separated by 30 min. For experiments involving citalopram (CTP, i.p., 10 mg/kg, Sigma Aldrich) treatment, pup littermates were injected intraperitoneally with vehicle (i.p., saline solution, once a day) or CTP from P2 to P14. For electrophysiological and behavioral experiments on CTP-treated animals, recordings for each day were carried out 24 h after the i.p. injection of the previous day, ensuring clearance of the drug from the animal on the day of the experiment[26–28]. For behavioral experiments in CTP-treated animals all pups in a litter were treated with either vehicle or CTP. Data shown in Fig. 2 and controls in Fig. 4 and Supplementary Fig. 5 comprise WT naive animals (3 litters) and vehicle-treated animals (5 litters), which were grouped together since they were not statistically different (two-way repeated measures ANOVA on ranks: $F_{(1,6)} = 4.77$, $P > 0.05$ for time spent together, $F_{(1,6)} = 0.27$, $P > 0.05$ for number of different clusters and $F_{(1,6)} = 0.77$, $P > 0.05$ for number of cluster switches. Two-way ANOVA on ranks: $F_{(1,44)} = 1$, $P > 0.05$ for primitive reflexes, $F_{(1,116)} = 2.87$, $P > 0.05$ for motor functions and $F_{(1,188)} = 2.38$, $P > 0.05$ for sensory functions.). Sample sizes were determined accordingly to the literature and previous experiments conducted in our laboratories. Number of animals and statistical analyses were derived to comply with the 3 R principle.

**Slice preparation for electrophysiology.** For acute slices, rat pups were anesthetized with isoflurane and transcardially perfused with an ice-cold cutting solution with the following composition (in mM): 115 NaCl; 3.5 KCl; 1.2 NaH$_2$PO$_4$; 25 D-Glucose; 25 NaHCO$_3$; 4 MgCl$_2$; 0.5 CaCl$_2$ (~300 mOsm, pH 7.4, oxygenated with 95% O$_2$ and 5% CO$_2$). The brain was removed and quickly immersed in the iced cutting solution. Coronal slices at the level of the somatosensory cortex (350 μm-thick; VT1000S, Leica Microsystems vibratome) were incubated at 35 °C for 30 min in an artificial cerebrospinal fluid (ACSF) with the following composition (in mM): 130 NaCl; 3.5 KCl; 1.25 NaH$_2$PO$_4$; 10 D-Glucose; 24 NaHCO$_3$; 2.5 CaCl$_2$; 1.5 MgCl$_2$ (~310 mOsm, pH 7.4, oxygenated with 95% O$_2$ and 5% CO$_2$). After 1 h recovery at room temperature (RT), slices were transferred to a recording chamber and perfused with ACSF (RT, 1.7 ml/min).

**Whole-cell patch-clamp electrophysiology.** Whole-cell voltage-clamp recordings were made from pyramidal neuron in Layer II/III, Layer V of the somatosensory cortex or CA1 of the hippocampus at RT. Glass micropipettes (resistance, 6–8 MΩ) were filled with an internal solution of (in mM): 140 KCl; 5 NaCl, 10 HEPES; 11 EGTA; 0.4 CaCl$_2$; 1 MgCl$_2$ (~300 mOsm, pH 7.28). Criteria for accepting a recording included an input resistance > 200 MΩ and a series resistance < 20 MΩ. Capacitance, input, and series resistance were measured online with Clampex 10.2 (Axon Instruments). For spontaneous postsynaptic current recordings, glutamatergic conductances were recorded by clamping cells at −70 mV, whereas GABAergic conductances were recorded by clamping the same cells at 0 mV. For miniature postsynaptic current recordings, cells were clamped at −70 mV: glutamatergic currents were isolated by bath application of tetrodotoxin (TTX, 1 μM, Tocris) and Bicuculline methiodide (BMI, 10 μM, Sigma Aldrich), whereas GABAergic currents were isolated from the same cell by bath application of TTX, 6,7-dinitroquinoxaline-2,3-dione (DNQX, 20 μM, Sigma Aldrich), and DL-2-amino-5-phosphonovaleric acid (APV, 50 μM, Sigma Aldrich). Each recording lasted for 10 min, with 5 min of recording in the presence of the first cocktail of drugs and 5 min of recording with the second cocktail of drugs. During the application of the second drug cocktail, the first 3 min were sufficient to wash out the previous drug from the slice. Data of sPSCs and mPSCs represent two exclusive sets of data. In experiments involving monitoring the efficacy of CNO in reducing spiking in slices electroporated with iDREADDs, cells were recorded in current-clamp mode. 500 ms square current pulses were injected into cells increasing in steps of 50 pA starting from −100 pA. Thereafter, CNO was bath perfused (0.5 mg/ml in 0.5% dimethyl sulfoxide, DMSO, Sigma Aldrich) for 5 min. mPSCs frequency of neurons from animals electroporated with iDREADDs or control vectors was observed before and after bath perfusion of 10 μM CNO. Data, filtered between 1 and 5 kHz and sampled at 25 kHz were acquired with a Multiclamp 700 B amplifier, and later analyzed using the pClamp 10.2 software (Axon Instruments). In Supplementary Fig. 1a, we classified low activity neurons those with a frequency below 0.17 Hz. This threshold value was calculated as the mean of the frequencies of both Glutamatergic and GABAergic postsynaptic currents between P2 and P8 (i.e. just before the abrupt increase in the frequencies of both currents were seen). Cells that did not show any mPSCs were excluded from the glutamate–GABA ratio analyses.

**DiI staining and spine counting.** Glass pipettes were tip filled with 2 μl DiI (DiI stock, 2 mg/ml in DMSO, Molecular Probes Inc.) and backfilled with pure DMSO. DiI was pressure-injected (500 nl. at 10 ms intervals) onto 300 μm acute brain slices cut as for electrophysiology and perfused with ACSF at RT (1.7 ml/min). After allowing the dye to diffuse for approximately 3 min, slices were fixed in 4% PFA overnight. On the next day, slices were washed several times with PBS and mounted with Vectashield (Vector Laboratories, Inc.) and examined with a Leica SP5 confocal microscope (Leica Microsystems). 50-μm-thick Z-stacks (0.8 μm step) were acquired with a ×63 oil-immersion objective (N.A. 1.4). Secondary dendrites of Layer II/III neurons were carefully marked in a 245 × 245 μm square (1024 × 1024 pixels resolution) and all focal planes were maximally projected. Spine density was calculated offline by counting the total number of spines and dividing it by the total length of the secondary dendrite on which they were counted using the measuring function of ImageJ 1.51.

**Golgi–Cox staining and spine counting.** Pups were perfused transcardially with 0.9% saline. Whole brains were dissected and immersed in the Golgi–Cox solution (5% potassium dichromate, 5% mercuric chloride, and 5% potassium chromate) for ~2–3 weeks. Then, brains were transferred to a 30% sucrose solution and stored in the dark at 4 °C. Coronal slices (100 μm-thick) were cut in 6% sucrose with a microtome (Leica SM200R sliding microtome) and transferred onto 2% gelatin-coated slides to initiate the staining process in humidified chambers. Ammonium hydroxide was applied for 30 min. Slices were then treated for 30 min with 1% sodium thiosulphate. Next, slices were treated with an increasing grade of ethanol starting from 50 to 100%, followed by a 15-minute immersion in solution X (1/3 parts by volume of chloroform, 1/3 parts by volume of xylene and 1/3 parts by volume of 100% ethanol). Slices were then treated with xylene for 15 min, mounted in mowiol, and examined under bright field with an upright Olympus BX51W microscope using a ×63 oil-immersion objective (N.A. 1.4). Spine density was calculated as for DiI staining.

**Immunohistochemistry.** Pups were transcardially perfused with 4% paraformaldehyde (PFA, Sigma Aldrich) in 0.1 M phosphate buffer (PB), pH 7.4. Brains were post fixed for 2 h at 4 °C, cryoprotected in 30% sucrose 0.1 M PB at 4 °C and cut coronally (80 μm-thick) with a freezing microtome (Thermo-Scientific). Slices were incubated in a citric acid-PBS solution for 30 min at 90 °C, allowing antigen retrieval. Then, slices were permeabilized and blocked with PBS containing 0.3% Triton X-100, 10% normal goat serum (NGS, Jackson Immuno Research) and 0.2% bovine serum albumin (BSA, Sigma Aldrich). Anti-SERT (rabbit polyclonal, #340003, 1:400, Synaptic Systems), anti-VGlut-1 (guinea pig polyclonal, #135304, 1: 2500, Synaptic Systems), anti-VGlut-2 (guinea pig polyclonal, #135404, 1: 1000, Synaptic Systems), anti-VGat (rabbit polyclonal, #131002, 1:700, Synaptic Systems), and anti-SERT antibodies were incubated in PBS containing 0.3% Triton X-100, 5% NGS, and 0.1% BSA. Immunostaining was detected using Alexa-568 and Alexa-488 fluorescent secondary antibodies (1:500, Invitrogen) prepared in the same blocking buffer as of the primary antibody. Sections were mounted in Vectashield (Vector Laboratories), examined with a Leica SP5 confocal microscope (Leica Microsystems) and processed on Adobe Photoshop (Adobe Systems). For immunohistochemistry on presynaptic proteins, brain slices were acquired using a ×63 oil-immersion objective (N.A. 1.4). Fifty μm-thick Z-stacks were acquired with a step size of 0.8 μm in a 245 × 245 μm square (1024 × 1024 resolution) positioned in Layer II/III of the somatosensory cortex. VGlut-1/2 and VGat puncta were counted offline using the 'analyze particles' function on ImageJ in an effective area (total area − area covered by cell nuclei) of the image and expressed as a ratio of absolute counts over the effective area. For high-magnification images of SERT-stained fibers in the somatosensory cortex, 60-μm-thick Z-stacks (0.8 μm step) were acquired with a ×63 oil-immersion objective (N.A. 1.4). SERT$^+$ fibers residing in a 245 × 245 μm square (1024 × 1024 pixels resolution) positioned in Layer II/III were counted on ImageJ after background subtraction using the 'Despeckle tool' and expressed as number of fibers per mm$^2$. Fiber density was measured in both hemispheres and in at least two different slices collected from the same pup; the values obtained were then averaged. For all experiments, brain slices were acquired and processed blind to postnatal age of the animal.

**Modified SHIRPA test and developmental index measurement.** Rat pups were subjected every day, from postnatal age 2 to 10, to 21 different motor and sensory tasks of SHIRPA (Supplementary Table 1). These tasks are a part of a battery of tests originally developed to evaluate neurological deficits in adult rodents. We modified the tests, making them suitable for analyzing behaviors in neonatal rats. An ~1-min interval was maintained between each task. For each experimental litter group (controls (WT plus vehicles), CTP, control (GFP) pre/postCNO, iDREADDs pre/postCNO, and control and iDREADDs mixed pre/postCNO) and for each type of behavior studied, we computed the percentage of animals that could perform that behavior at a particular postnatal day (developmental index). In order to find groups of functions with similar developmental profiles, we then performed a principal component analysis (PCA) followed by a k-means clustering of '2' on the time course of each developmental score. The analysis individuated two different behavioral clusters, which we defined as: primitive reflexes (fully developed already at P2), and motor/sensory functions (mostly developing between P6 and P8). We further split the motor/sensory classification in two different classes (motor and sensory) based on their phenotypic domain. Developmental indexes for each of the three behavioral classes (primitive reflexes, motor functions and sensory functions) were computed as the mean ± SEM for all behaviors in that particular class, as indicated in Supplementary Table 1. Polar plots are bar graphs quantifying the percentage of animals that could perform a behavior plotted in polar coordinates.

**Huddling test.** For huddling behavior, each litter was formed by 10 pups and a dam. Littermates were isolated from their mother and introduced in an empty arena (40 cm × 40 cm) where they were all separated from one another by ~9.4 cm on the circumference of a circle (30 cm diameter) drawn on the floor of the arena. Ten-minute videos (camera: Canon XF105 HD Camcorder, Canon; Sony HXR-NC2500 AVCHD Camcorder, Sony Corporation) of freely moving pups were recorded each day starting from P2 to P10 always at 10 AM. From the recordings, we extracted one frame every 30 s and visually scored the behavior of all pups in huddling groups based on their proximity and interaction. We considered a pup doing huddling when it made active and prolonged contact with one or more littermates by using his head, snout or when he formed a pile with them (Supplementary movie 1). Based on this scoring, we extracted three measures of interaction, namely 'time spent together', 'no. of different clusters', 'no. of cluster switches' (Supplementary Table 2; Supplementary movie 2 as an example of a single cluster switch). Huddling was performed randomizing the order of control, iDREADDs and control and iDREADDs litters within each experimental session. Videos were then processed by extracting one frame every 30 s and mouse position was manually tracked using ImageJ package MTrackJ. Every image was then divided in squares of 18 cm$^2$ and the total space occupied was calculated as the total number of squares in the arena visited by each pup during the huddling experiment (Supplementary Table 2). For temperature drop, we recorded the core body temperature of rat pups during the huddling task with an infrared thermocamera (FLIR Systems). Only for animals that never huddled during the recording period, we calculated the difference in the body temperature between the first and the last

frame of the recording (Supplementary Figs. 5b, d). All behavioral tests and data analysis were performed with the experimenter blind of the experimental group.

**iDREADD construct and in utero electroporation**. iDREADD construct. A full-length coding sequence of hM4Di was obtained by performing a PCR of the plasmid pcDNA5/FRT-HA-hM4DGi (Addgene), using two synthetic oligonu-cleotide primers (5′- TAACAGCGGCCGCATGTACCCATACGATGTTCC-3′ and 5′- TCAGTGACTCGAGCTACCTGGCAGTGCC-3′) with restriction to the 5′ end of each primer. The Not1–XhoI fragment of the PCR product was then sub-cloned into the pCAGGs-IRES-eGFP.

In utero electroporation of the somatosensory[15,16,48]. Timed-pregnant Sprague Dawley (SD, Charles River) rats were anesthetized at E17.5 with isoflurane (induction, 3.5%; surgery, 2.5%), and uterine horns were exposed by laparotomy. Control (pCAG-IRES-eGFP) and inhibitory DREADD (pCAGG-hM4D(Gi)-IRES-eGFP) vectors (2.0 µg/µl in water) together with the dye Fast Green (0.3 mg/ml; Sigma Aldrich) were injected (5–6 µl) through the uterine wall into one of the embryo's lateral ventricles by a 30-gauge needle. After the injection, the embryo's head was placed between tweezer-type circular electrodes (10-mm diameter; Nepa Gene). For the electroporation protocol, we applied 5 electrical pulses (amplitude, 50 V; duration, 50 ms; intervals, 150 ms) delivered with a square-wave electroporation generator (CUY21EDIT; Nepa Gene). Uterine horns were returned into the abdominal cavity, and embryos continued their physiological development. All iDREADDs litters were electroporated with inhibitory DREADDs (median electroporation efficiency 90%), while control litters with the same concentration of GFP (median electroporation efficiency 80%). For the control and iDREADDs mixed litters, about half of the pups were transfected with iDREADDs and half with control vector (median electroporation efficiency for iDREDDS 40%).

**Statistics**. Statistical analysis was performed with R, version 3.3.1, and Sigma Plot 11.0. Average data were represented as mean ± SEM. Equal distribution of variances and normal distribution of the residues were inspected by Levene's and Shapiro–Wilk test, respectively; if a violation of parametric tests assumptions was detected, the corresponding non-parametric test was executed. Precisely, pairwise comparisons were performed by using t-test or a Wilcoxon rank-sum test; when two samples were paired, a paired t-test or a Wilcoxon-signed rank test were run. When more than two groups were analyzed, one-way ANOVA followed by pairwise t-test comparisons with Holm's correction was used by default; in case of heteroscedastic data, a type II ANOVA with 'h3' correction was preferred. When both parametric tests assumptions were violated, a Kruskal–Wallis test followed by Dunn's post hoc test corrected with Holm's method was used. If two factors were present, a two-way ANOVA on normal or ranked values was employed, depending on the violation of parametric tests assumptions. The Pearson's coefficient of correlation was calculated from the linear regression analysis. All statistical tests were two-tailed except where specified otherwise. Minimum level of significance was set to $P < 0.05$.

## Code availability

The Python script used for calculating 'Time spent together', 'No. of different clusters', and 'No. of cluster switches' is available alongside a minimum working example of scoring at http://gitlab.iit.it/lcancedda/huddling-scripts.git.

## Data availability

The data that support the findings of this work are available from the authors upon request.

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

## Acknowledgements

We thank Annalisa Savardi, IIT for designing the inhibitory DREADD clone. We also thank Duilio Farina and Nicholas Dring, IIT for their assistance with graphics and image processing, and Smitabh Moitra for his help with animal tracking in ImageJ. This work was partially funded by the European Research Council (ERC) under the European Union's Horizon 2020 research and innovation program (Grant Agreement No. 725563 to L.C.) and Telethon Foundation (GGP13187 and TCP15021 to L.C.).

## Author contributions

S.N. carried out all electrophysiological, behavioral and anatomical experiments, analyzed the data, and wrote the manuscript. E.B. and R.N. analyzed behavioral data and wrote parts of the manuscript. R.N. performed part of the histological, behavioral and in utero electroporation experiments, and analyzed the data. A.W.C. performed part of the in utero electroporation experiments. V.T. designed and supervised all behavioral experiments and wrote the manuscript. L.C. designed the electrophysiological, histological and behavioral experiments, analyzed behavioral data, and wrote the manuscript. All of the authors read and commented on the manuscript.

## Additional information

**Competing interests:** The authors declare no competing interests.

