## [Peer Review File · Nature Communications]

Reviewers' comments:

Reviewer #1 (Remarks to the Author):

In this manuscript the authors examine the postnatal developmental time-course of GABAergic and glutamatergic synapse formation in rat somato-sensory cortex and the parallel appearance of primitive behaviors related to reflex, sensory, and motor system function. They find that GABAergic and glutamatergic synapse formation occurs in a parallel, coordinated fashion, unlike some reports from hippocampus where GABAergic precedes glutamatergic synapse formation. Synapse formation is monitored both morphologically by counting spine and bouton density as well as electrophysiologically by recording spontaneous and mini PSCs across ages. They also document that while primitive reflex responses are evident at birth, sensory and motor responses appear at P7 and huddling behavior appears at P9, the latter corresponding to a large increase in synapse formation found in their electrophysiological study.

In the second part of the manuscript the authors test the causal relationship between cortical neural activity and the appearance of primitive behaviors. They use the CNO/hM4D pharmacogenetic system following expression of hM4D by in utero electroporation into the superficial somatosensory cortex followed by CNO delivery to inhibit action potential firing and examine the expression of behaviors in P9 pups. They find that reflex, motor, and sensory responses are unaffected, but that huddling behavior decreases, suggesting that somatosensory cortex neural activity sustains huddling at this age. They then test the hypothesis that extracellular serotonin (previously shown to affect postnatal neural circuit development) is important for timing the appearance of both synapse formation and behavior during this period by systemic treatment with the selective serotonin reuptake inhibitory citalopram from P2-P15. Citalopram treatment was associated with precocious appearance of mEPSCs and mIPSCs and huddling, but not motor and sensory behaviors.

This study presents a large body of work that is an important addition to the surprisingly poorly understood field of early cortical circuit development. In particular, the finding of closely parallel GABAergic and glutamatergic synapses development in cortex points out how dramatic and abruptly synaptogenesis is in these structures and how early behavioral milestones are achieved in the absence of cortical synapses. Generally, the data appear sound and of high quality.

Comments:

1. The CNO experiments need controls. The electrophysiology has a no-hM4D control, but the behavioral tests do not. These must be done, especially as the dose of CNO (10 mg/kg) is relatively high.

2. The manuscript is poorly written and must be thoroughly re-worked. Nearly every sentence suffers from usage and grammatical errors and many phrases remain completely uninterpretable and ill-defined. Some are listed here as examples:

- The sentence "While most motor, sensory and survival-permissive pup behavior were..." appears out of place – better to first say what it is associated with and then the causal evidence.
- "purposive" is not used correctly.
- The intro sentence "Brain development follows specific..." is too general, obvious, and vague.
- The word "indeed" is incorrectly used in "...sequence for synaptogenesis was indeed reported with ..."
- "sequences of events" is incorrect.
- "unlock quick and safe developmental programs" appears colloquial and is not defined.
- The verb in "...neuronal networks... start to code for both primitive..." is incorrectly used.
- The verb in "brain structures that seats the highest executive function" is incorrectly used.
- The word "difference" is incorrectly used in "the frequency of spontaneous GABAergic currents reached a significant difference only later..."
- The word "behavior" is used incorrectly in "...emphasizing the anticipated behavior of glutamate

compared to GABA.”

- The word “reliability” is used incorrectly in “...highlights the reliability of our recording configuration...”
- Etc.

Reviewer #2 (Remarks to the Author):

In their manuscript, Naskar and colleagues analyze the sequential development of functional and behaviorally relevant synapses in the upper cortical layers of rat. By combining electrophysiology, morphological analysis and behavioral testing the authors reveal interesting new findings and based on their data propose a ‘ready’, ‘steady’, ‘go’ model for neocortical development. Overall the manuscript is quite timely, well prepared and the data is conclusive. The model which the authors propose is intriguing and potentially of broad interest to the neuroscience community. In order to solidify some of the major claims the following points should be addressed:

1. The authors show nicely that within just two days the frequency of both glutamatergic and GABAergic spontaneous currents increases significantly. They then analyze the morphological correlates and found that spine density and synaptic profiles (by virtue of marker expression) in layer II/III also increased from P7 to P9. Since pruning of synapses during the early postnatal stages significantly contributes to the emergence of synaptic specificity it would be important to also analyze later time points. Could it be that the increase in synapses is just a temporary feature? Alternatively, even with pruning the density of synapses may even further increase at later stages and potentially shape integrative behavior (such as huddling) beyond the period studied here.
2. The morphological analysis of spine density of layer II/III neurons is conclusive as presented but the authors also mention that temporal development of synaptogenesis occurs in a different fashion among distinct cortical layers. While the electrophysiology data is convincing (upper vs lower layer properties) it would be important to show potential morphological differences (upper vs lower layer) as well.
3. The model as presented in Figure S7 is illustrative but the key message of the paper could be reflected a bit better. In effect, the authors claim a very important temporal component in layer II/III synaptic development but this does not really become apparent in the model. As presented the model rather focuses on the development of thalamocortical synapses and the involvement of serotonin.
4. The authors could extend the introduction to more comprehensively discuss the state of the current literature. For the general reader some more background may help to fully appreciate the conceptual advance of the study.
5. Related to the above point, it is quite intriguing how the temporal sequence of synapse development matches the emergence of reflexes, simple behavior and eventually more integrated behavior and social interaction. To what extent do these new findings apply to other rodents such as the mouse with a much smaller brain? In other words, do the new findings reflect properties in neuronal circuit assembly that are specific to rats (with more complex brain than mouse), or is the concept of temporally controlled synapse development a more general principle instructing the emergence of behavior? The authors may want to elaborate a bit further on these aspects in the discussion.

Reviewer #3 (Remarks to the Author):

This manuscript describes a role for the cerebral cortex in huddling behavior in early postnatal rat pups. The authors link the emergence of this behavior to the dynamic maturation of synaptic connectivity in layer 2/3 of the somatosensory cortex. In my opinion, the connection between the synaptic analyses and the behavioral experiments is limited. Moreover, the main results — related to the huddling behavior — are interesting but preliminary. Finally, the manuscript lacks clarity and detail in several important passages, as detailed below.

Main issues

The analysis of the quantification of huddling behavior is insufficiently described. It is unclear what the terms “Time spent together”, “No. of different clusters” and “No. of cluster switches” exactly means. The authors seem to be using these as a description of a collective behavior, when one would like to know these variables for each of the individual pups. As for the collective analyses, it is difficult to understand why the number of clusters increases over time, when one would predict that huddling would actually reduce the final number of clusters (i.e. with most pups together in one or two groups as shown in Figure 2C). The authors should do a better job at explaining these measurements in the text.

The interpretation of the CNO experiments is difficult, and this is linked to my previous comments. It seems that the behavior has been estimated collectively, but appropriate controls are missing. First, the authors should look at litters of electroporated pups that are then injected with vehicle, and measure huddling before and after. Most importantly, the authors should perform experiments in which for example only some pups are treated with CNO in the litter, while littermate are injected with vehicle. Does the huddling behavior differ for these pups?

While the treatment with Citalopram can be linked to changes in the development of excitatory and inhibitory synapses in the cortex and in huddling behavior, there is not experiment connecting both sets of experiments. In other words, Citalopram may affect huddling behavior through a mechanism that does not involve synaptic changes in the somatosensory cortex.

It is difficult to understand how the reported changes in layer 2/3 somatosensory cortex might be linked to decreased huddling. What is the authors hypothesis?

Point by point response to the reviewers' comments

In the revised version of the manuscript, we highlighted in red the corrections to the former version to simplify the reviewing process.

Reviewer #1 (reviewer's comments are in italics):

We are glad that the reviewer thinks that *“this study presents a large body of work that is an important addition to the surprisingly poorly understood field of early cortical circuit development”* and that *“ Generally, the data appear sound and of high quality”*.

Our responses to the reviewer's comments are as it follows:

1) The CNO experiments need controls. The electrophysiology has a no-hM4D control, but the behavioral tests do not. These must be done, especially as the dose of CNO (10 mg/kg) is relatively high.

We thank the reviewer for raising this point. We have now done appropriate controls for our DREADDs experiments with a no-hM4D control (i.e. transfection of control vector, GFP only), as suggested by the referee. We found no significant alterations in the three huddling parameters inspected after CNO injection. The new results are now described in the manuscript on page 8 and represented in Fig. 3F-G. We acknowledge that the dose of CNO is relatively high, when compared to standard doses in mice (i.e., 1-3 mg/kg, but see also Ray et al.¹, where they used a 10 mg/kg dose of CNO). Nevertheless, in rats, there have been studies using systemic administration of CNO at a dosage of 5-10 mg/kg of CNO (reviewed in MacLaren et al.²). So, a 10 mg/kg dose of CNO can be considered in a range of standard dosage for experiments in rats, as for our case.

2) The manuscript is poorly written and must be thoroughly re-worked. Nearly every sentence suffers from usage and grammatical errors and many phrases remain completely uninterpretable and ill-defined.

We thank the reviewer for the thorough reading of the manuscript. The text of the revised manuscript has been now proof read by a professional company (American Journal Experts, see the certificate that we uploaded as an attachment to the manuscript).

Moreover, we have also specifically addressed each of the example grammatical issues raised by the reviewer below:

- *The sentence “While most motor, sensory and survival-permissive pup behavior were...” appears out of place – better to first say what it is associated with and then the causal evidence.*

The sentence has been fixed and it now reads: *“while most motor and sensory behaviors, which are fundamental for pup survival, were already in place at approximately P7”* (page 1)

- *“purposive” is not used correctly.*

The word “purposive” has been replaced by *“integrative huddling behavior”* or *“socially directed behaviors”* or *“social, integrative behavior”* throughout the manuscript (page 1, 6) and removed from the sentence *“which require reflex responses but purposive movement”* that now reads *“which require reflex responses but little movement”* (page 12).

- *The intro sentence “Brain development follows specific...” is too general, obvious, and vague.*

We have erased the vague sentence altogether and slightly modified the following sentence *“In particular, the last decade of investigations in neurodevelopment has provided a general model of how timed sequences of events occur at cellular and network levels in a similar fashion across diverse brain areas.”* in *“Investigations in developmental neurophysiology performed over the last decade have provided a general model of how timely, sequential events occur at cellular and network levels in a similar manner across diverse brain areas.”* (page 2).

• *The word “indeed” is incorrectly used in “...sequence for synaptogenesis was indeed reported with ...”*
The word “indeed” has been erased from that sentence (page 2).

• *“sequences of events” is incorrect.*

The phrase “sequences of events” has been erased by the manuscript and replaced by more appropriate expressions such as “timely, sequential events” and “the timing and mechanisms” (page 2 and 11)

• *“unlock quick and safe developmental programs” appears colloquial and is not defined.*

We have rephrased the whole paragraph and better defined the concept. The new paragraph now reads “The depolarizing actions of GABA and early synchronous activity during the first postnatal week are pivotal for the morphological and functional maturation of neurons and the establishment of their first connections” (page 2)

• *The verb in “...neuronal networks... start to code for both primitive...” is incorrectly used.*

We have rephrased in “Then, the initial connections mature ... and their finely tuned activity begins to encode both primitive complex behaviors (e.g., reflexes, sensory and motor functions) and subsequent integrative behaviors (e.g., social and cognitive)” (page 2).

• *The verb in “brain structures that seats the highest executive function” is incorrectly used.*

We have replaced the word “seats” with the word “controls”.

• *The word “difference” is incorrectly used in “the frequency of spontaneous GABAergic currents reached a significant difference only later...”*

We have rephrased the sentence that now reads “Conversely, the frequency of spontaneous GABAergic currents only was significantly different from 0 only later, at P7” (page 3).

• *The word “behavior” is used incorrectly in “...emphasizing the anticipated behavior of glutamate compared to GABA.”*

We have rephrased the sentence that now reads “...emphasizing the fact that the glutamatergic conductance appeared earlier than the GABAergic conductance” (page 4).

• *The word “reliability” is used incorrectly in “...highlights the reliability of our recording configuration...”*

We have erased the sentence altogether.

Reviewer #2 (reviewer's comments are in italics):

We are glad that the reviewer thinks that *“The manuscript is quite timely, well prepared and the data is conclusive.”* and that *“The model which the authors propose is intriguing and potentially of broad interest to the neuroscience community.”*

Our responses to the reviewer's comments are as it follows:

1) The authors show nicely that within just two days the frequency of both glutamatergic and GABAergic spontaneous currents increases significantly. They then analyze the morphological correlates and found that spine density and synaptic profiles (by virtue of marker expression) in layer II/III also increased from P7 to P9. Since pruning of synapses during the early postnatal stages significantly contributes to the emergence of synaptic specificity it would be important to also analyze later time points. Could it be that the increase in synapses is just a temporary feature? Alternatively, even with pruning the density of synapses may even further increase at later stages and potentially shape integrative behavior (such as huddling) beyond the period studied here.

We thank the reviewer for raising this question. We have now extended our electrophysiological recordings, and behavioural assessment and morphology analysis to later points. In particular, although the synaptogenesis plateaued already at around P15, we performed electrophysiological recordings also at P30. We found that mPSCs frequency

did not further increase (glutamatergic mPSCs: 4.28 ± 0.57 Hz at P15 vs 3.74 ± 0.39 Hz at P30; GABAergic mPSCs: 1.92 ± 0.46 Hz at P15 vs 2.22 ± 0.44 Hz at P30). The new results are shown below in Figure 1 for the reviewer.

Figure 1 for the reviewer. Development of glutamatergic and GABAergic miniature postsynaptic currents (mPSCs) of Layer II/III pyramidal neurons from P2 to P30. Dots indicate the average \pm SEM of mPSCs recorded at a single postnatal day normalized to the mean recorded at P15 ($N = 164$ neurons from 56 animals). No significant difference was found between P15 and P30 mPSCs frequency, Kruskal-Wallis test, *post hoc* Dunn's test with Holm's correction.

We paralleled the electrophysiological data with additional experiments on huddling behavior at stages later than P10. Nevertheless, already by P11-15 the motility of the mice and their huddling strategy changed quantitatively and qualitatively so much that it made it impossible to make a direct comparison with the huddling measures that we presented in our manuscript (see Supplementary video for the reviewer uploaded as an extra file together with the revised manuscript). It appears obvious that huddling later in life should not be parameterised as in young individuals to maintain a proper construct validity. Indeed, we are currently working on a new manuscript in which we will describe and analyse the new type of huddling behavior that characterize older animals. New sets of parameters are under investigation to quantify huddling in older animals.

Thus, although the electrophysiological and behavioural studies at later stages that we reported above were not included in the revised manuscript, we still expanded also the analysis of Layer II/III spine density until P15, for completeness of Fig. 1. We have found no significant increase in comparison to P9 (Fig 1D). This is in line with the electrophysiological data of mPSCs frequency of upper layer neurons (Fig 1B).

2) The morphological analysis of spine density of layer II/III neurons is conclusive as presented but the authors also mention that temporal development of synaptogenesis occurs in a different fashion among distinct cortical layers. While the electrophysiology data is convincing (upper vs lower layer properties) it would be important to show potential morphological differences (upper vs lower layer) as well.

We thank the reviewer for raising this point. We have now carried out a new experiment and analyzed spine counts of deep layer neurons at P7, P8, P9, P10 and P15. In line with electrophysiological data, our results show that the spinogenesis profile of layer V increases progressively from P8 to P15. The new results are now represented in Supplementary Fig. 2 C and D, described in the manuscript on page 5, and discussed on page 11.

3) The model as presented in Figure S7 is illustrative but the key message of the paper could be reflected a bit better. In effect, the authors claim a very important temporal component in layer II/III synaptic development but this does not really become apparent in the model. As presented the model rather focuses on the development of thalamocortical synapses and the involvement of serotonin.

We thank the reviewer for this comment. We have now modified Supplementary Fig. 7 (now Supplementary Fig. 8) so as to depict the key message of the manuscript that reflects on the temporal scale of synaptic development in layer II/III (we depicted a larger increase of synaptic connections at P7-9) and not only the development of thalamocortical synapses. We also exemplified the development of complex behaviors with cartoons of rat pups for the different ages (Supplementary Fig. 8).

4) The authors could extend the introduction to more comprehensively discuss the state of the current literature. For the general reader some more background may help to fully appreciate the conceptual advance of the study.

We have now modified the introduction and have discussed literature on the developmental sequence of GABA vs glutamate synaptogenesis and spontaneous patterns of neuronal activity in different brain areas (page 2 of the manuscript) to be able to give some background to the general reader, as suggested by the reviewer.

5) Related to the above point, it is quite intriguing how the temporal sequence of synapse development matches the emergence of reflexes, simple behavior and eventually more integrated behavior and social interaction. To what extent do these new findings apply to other rodents such as the mouse with a much smaller brain? In other words, do the new findings reflect properties in neuronal circuit assembly that are specific to rats (with more complex brain than mouse), or is the concept of temporally controlled synapse development a more general principle instructing the emergence of behavior? The authors may want to elaborate a bit further on these aspects in the discussion.

We thank the reviewer for raising this point. We have now added a paragraph in the discussion expanding on this aspect. The new paragraph reads “Huddling behavior has been observed in 67 different mammalian species, 40 of which are rodents. Thus, the findings of the present study can be reasonably generalized to at least these models, assuming that a similar developmental timeframe of synaptogenesis affects altricial, social behaviors. Our study points out that the overall maturation of neuronal circuits is a temporally controlled process that exerts a profound influence over early integrative behaviors. For example, GABAergic synaptic maturation shapes the sensory integration of the mouse insular cortex (IC), a region involved in emotional and cognitive functions. In particular, in a mouse model manifesting social deficits, inhibitory neurons showed impaired postnatal maturation and weakened synaptic transmission. IC sensory integration was restored after neuronal GABA_A receptor activity was increased through the injection of a positive allosteric modulator, but only during an early time window corresponding to circuit development. This finding highlights the possible concomitancy between correct synaptic maturation and the emergence of social behaviors.” (page 13 and 14 of the manuscript).

Reviewer #3 (*reviewer's comments are in italics*):

We are glad that the reviewer thinks that “*The main results — related to the huddling behavior — are interesting*” .

Our responses to the reviewer's comments are as it follows:

1) The analysis of the quantification of huddling behavior is insufficiently described. It is unclear what the terms “Time spent together”, “No. of different clusters” and “No. of cluster switches” exactly means. The authors should do a better job at explaining these measurements in the text.

We apologize for all missing information in describing behavioral measures. We have now clarified these parameters in the supplementary section of the manuscript (Supplementary Table 2).

In particular, we defined:

Time spent together as the average time, expressed in minutes, that each pup spends forming a cluster with every other littermate during the entire huddling session (10 minutes). The individual measurements of all pups belonging to a same litter are then averaged to obtain a value representative of the corresponding litter.

No. of different clusters as the number of clusters formed by a unique combination of any number of pups during the 10 minutes huddling session.

Although formed in different timepoints, two or more clusters composed of the same combination of pups are considered as a same cluster.

No. of cluster switches as the number of times a pup switched from a cluster to another one between two consecutive sampling intervals of 30 seconds each. For each litter, the sum of all the clusters formed by the pups during the huddling session was considered.

Moreover, we have included one video representative of the typical huddling behavior and one representative of an episode of cluster switching for further clarification (Supplementary video 1 and 2). Both videos were recorded at P9).

The authors seem to be using these as a description of a collective behavior, when one would like to know these variables for each of the individual pups.

We apologize for the misunderstanding. Most of the behavioral measures refer to individuals.

For example, *Time spent together* & *No. of cluster switches* are measurements that are derived from scoring the individual behavior of each single pup (as indicated in Supplementary Table 2, last column). Therefore, for each litter, averaged single-pup values of Time spent together and summed single-pup values of No. of cluster switches are reported (Supplementary Table 2, middle column). To address the reviewer's concern, we have now analyzed the data also as an average of all animals independent of the litter they belong to. Grouping for litter (Fig. 2D) or analyzing data points from single pups all together (Fig. 2 for the reviewer), does not change the main message of the manuscript, which is that between P8-9 there is an abrupt increase of the huddling parameters.

Figure 2 for the reviewer. Quantification of the average values \pm SEM for the two individual huddling parameters (Time spent together and No. of cluster switches) between P8 and P9. Each dot represents a single pup value. Data for single pups were derived from the same litters analyzed in Fig. 2D, ($N = 80$ pups, from 8 litters). Both tests were performed on the same set of animals (pups not performing cluster switches at P8 or P9 were removed from the graph on the right); Wilcoxon signed rank test. *** $P < 0.001$.

However, we did include a collective parameter in our analysis: number of different cluster (Supplementary Table 2, last column), which is the number of clusters formed by a unique combination of any number of pups during the 10 minutes huddling session. By definition, this is a group measurement. Indeed, different pups may take part of the same cluster at one specific point in time and of many different others at one other point in time.

As for the collective analyses, it is difficult to understand why the number of clusters increases over time, when one would predict that huddling would actually reduce the final number of clusters (i.e. with most pups together in one or two groups as shown in Figure 2C).

We apologize with the reviewer for the misunderstanding. The parameter that we quantify is not number of *final* clusters, but number of *different* clusters. As now reported in Supplementary Table 2, the Number of different clusters is defined as the number of clusters formed by a unique combination of any number of pups during the 10 minute huddling session. Thus, with time, when the pups increase their social behavior and the number of cluster switching that they perform, it is intuitive to understand that the number of different clusters increases.

2) The interpretation of the CNO experiments is difficult, and this is linked to my previous comments. It seems that the behavior has been estimated collectively, but appropriate controls are missing. First, the authors should look at litters of electroporated pups that are then injected with vehicle, and measure huddling before and after.

We understand the concerns/confusion of the reviewer and we hope that we have now solved the misunderstanding within the last few points. We discussed the issue of the control experiment with the editor who instructed us that the control experiments to be added to the manuscript were the same as for reviewer 1's comment. Thus, we have performed other control experiments with transfection of control vector (GFP only) and quantification of huddling behavior before and after CNO treatment. After CNO injection, we found that all the three huddling parameters, as well as the developmental index, did not change in comparison with the values observed before the injection. The new results are now described in the manuscript on page 8 and 9 and represented in Fig. 3F-G.

Most importantly, the authors should perform experiments in which for example only some pups are treated with CNO in the litter, while littermate are injected with vehicle. Does the huddling behavior differ for these pups?

We apologize with the reviewer for the confusion. The experiments described in new Figure 3L-M (old Figure 3E-F) are indeed experiments on mixed litters with some pups electroporated with iDREADDs and their littermates electroporated with control vector (GFP only), and assessed before and after CNO treatment. We have now made much clearer this in the text on page 7 and 8 of the main manuscript and Figure 3L-M. In line with comment 1 of the referee above, we have now analyzed for single animals all the litters where both Control and iDREADDs pups were present in the same litter. We have thus averaged all DREADDs-electroporated pups together or all control vector-electroporated pups together independent of their original litter. In particular, we quantified the two individual-pup parameters ("Time spent together" and the Number of cluster switching") and found them significantly reduced upon CNO treatment in both iDREADDs and Control groups (Supplementary Fig. 6C-D). This is not surprising, as it is in line with the fact that huddling is a group behavior (see comment 1 for referee 3). Being part of a litter where pups with impaired huddling behavior are present (iDREADDs-transfected pups) will for sure affect the huddling behavior of the control littermate pups during their common huddling session. This is now discussed on page 8,9 of the revised manuscript.

Thus, to address the reviewer's concern and specifically address huddling in the presence of cortical inhibition by DREADDs, we performed new experiments where we electroporated the entire litter with iDREADDs and assessed huddling before and after CNO treatment. As controls this time, we utilized litters where all pups were electroporated with control vectors (GFP only) and assessed for huddling before and after CNO treatment. We found

that litters transfected with iDREADDs showed impaired huddling behavior in comparison to Control litters (which showed comparable huddling behaviors before and after CNO treatment), and no alteration of developmental index. These new results have been described in the manuscript (pages 7, 8 and 9) and represented in Fig. 3F-I.

A direct comparison among all the huddling behavior groups is now reported in Supplementary Fig. 6.

3) While the treatment with Citalopram can be linked to changes in the development of excitatory and inhibitory synapses in the cortex and in huddling behavior, there is not experiment connecting both sets of experiments. In other words, Citalopram may affect huddling behavior through a mechanism that does not involve synaptic changes in the somatosensory cortex.

We agree with the referee, as we are aware that citalopram may affect huddling behavior by some other mechanism. Indeed, this is always the case when trying to assess causality with a pharmacological approach. In the manuscript, we had been very careful not to claim any causality between the effect of citalopram on synaptic mPSCs and huddling. Moreover, we designed and performed the iDREADDs experiments to at least causally link activity in the somatosensory cortex and huddling behavior. To try to strengthen our point, we have now included new experiments addressing the spontaneous miniature synaptic events (mPSCs) in animals transfected with iDREADDs and control vectors. Consistent with our hypothesis, we found a significant decrease of mPSCs frequency only in iDREADDs-transfected animals. The new analysis has been described in the text on page 7,8 and represented in Fig 3E. Moreover, as pointed out by the referee, we have now also made explicit the fact that Citalopram may affect huddling behavior through a mechanism that does not involve synaptic changes in the somatosensory cortex in the discussion section of the manuscript (page 12,13).

4) It is difficult to understand how the reported changes in layer 2/3 somatosensory cortex might be linked to decreased huddling. What is the authors hypothesis?

Huddling, as we define it in our manuscript, is a social behavior. Within the neocortex, social behaviors have their neural representation in the limbic allocortex, particularly in the entorhinal-perirhinal region³. The somatosensory cortex receives and sends prominent reciprocal excitatory inputs with the entorhinal-perirhinal region that converge in the supragranular/superficial layers⁴. Our electroporation strategy ensured specific targeting iDREADDs to Layer II/III neurons, and thus manipulation of neural transmission in this cohort of neurons. Our hypothesis is that decreasing neural transmission in the superficial layers of the somatosensory cortex by CNO infusion, also decreases activity in the subset of Layer II/III neurons that project to the entorhinal-perirhinal region, thus possibly affecting huddling behavior. We have added a new paragraph in the discussion session addressing this issue (page 12,13).

Bibliography

1. Ray, R. S. *et al.* Impaired respiratory and body temperature control upon acute serotonergic neuron inhibition. *Science* (80-.). **333**, 637–642 (2011).
2. MacLaren, D. A. A. *et al.* Clozapine N-Oxide Administration Produces Behavioral Effects in Long-Evans Rats: Implications for Designing DREADD Experiments. *eNeuro* **3**, (2016).
3. Talamini, L. M., Koch, T., Luiten, P. G., Koolhaas, J. M. & Korf, J. Interruptions of early cortical development affect limbic association areas and social behaviour in rats; possible relevance for neurodevelopmental disorders. *Brain Res* **847**, 105–120 (1999).
4. Aronoff, R. *et al.* Long-range connectivity of mouse primary somatosensory barrel cortex. *Eur J Neurosci* **31**, 2221–2233 (2010).

REVIEWERS' COMMENTS:

Reviewer #1 (Remarks to the Author):

The authors have satisfied my concerns over the general presentation of their work as well as the important addition of IDREADD controls

Reviewer #2 (Remarks to the Author):

The authors now provide compelling responses to all major issues that were raised in the initial review of this manuscript. The authors added new data which greatly strengthen the main conclusions of the manuscript. In my opinion, this manuscript adds significantly to our understanding of the emergence of neocortical microcircuitry and behavior. Overall this study is likely to be of great interest to the broader readership. Therefore I would suggest that the beautiful and very informative model as presented in the supplementary data (supplementary figure 8) is highlighted better and moved to the main manuscript as Figure 5.

Reviewer #3 (Remarks to the Author):

The authors have done a good job at revising the manuscript in response to my queries and those from the other two reviewers. The control CNO experiments are a welcome addition, as are the clarifications around the terms used in the analysis of huddling. The Citalopram continues to be a dirty experiment that, as recognized by the authors, does not link well with their findings in the somatosensory cortex. This drug was administered via ip, and so it could affect serotonin function essentially anywhere in the brain. From an experimental point of view, this remains a weak observation. The authors could have used a number of other approaches to specifically test the role of serotonin in the cortex in the context of huddling behavior.

Linking neural circuits to behavior is not trivial, and so the authors are to be commended for their efforts. That being said, the authors exhibit a rather unnecessary tendency to overstretch their interpretations. The entire discussion around the idea of the 'steady, ready, go' organization of the cortex (pages 11 and 12) is purely speculative and is not based on experimental evidence. There is not a single piece of evidence in the paper suggesting that the somatosensory cortex is required for feeding during early postnatal development, for example. The suggestion that projections from the somatosensory cortex to limbic regions of the cortex mediate the function of S1 in huddling is also speculative and could have been tested experimentally.

Point by point response to the reviewers' comments

Reviewer #2 (*reviewer's comments are in italics*):

- 1) *Therefore I would suggest that the beautiful and very informative model as presented in the supplementary data (supplementary figure 8) is highlighted better and moved to the main manuscript as Figure 5.*

We thank the reviewer for his suggestion. However, since reviewer #3 has expressed concerns about the overall 'steady, ready, go' model for the cortex organization during development (see points number 2-4 of referee #3), we decided to keep the corresponding figure in the Supplementary Information file.

Reviewer #3 (*reviewer's comments are in italics*):

- 1) *The Citalopram continues to be a dirty experiment that, as recognized by the authors, does not link well with their findings in the somatosensory cortex. This drug was administered via ip, and so it could affect serotonin function essentially anywhere in the brain. From an experimental point of view, this remains a weak observation. The authors could have used a number of other approaches to specifically test the role of serotonin in the cortex in the context of huddling behavior.*

We agree with the reviewer that our Citalopram experiments illustrating the role of 5HT neurotransmission are not entirely clean because inhibiting 5HT reuptake by Citalopram affects 5HT transmission anywhere in the brain. Our hypothesis is that 5HT reuptake by SERT in the thalamo-cortical (TC) afferents during the first week of postnatal life modulates huddling. 5HT transmission in the thalamus is dominated by the 5HT1 group of metabotropic receptors (particularly, 5HT1A, 5HT1B, 5HT1D and 5HT1F¹). In order to delineate the role played by 5HT transmission in regulating huddling specifically in the first week of postnatal life, the cleanest alternative would have been to knock down each 5HT1 receptor subtype in TC afferents using SERTfl/+1;VGlut2-Cre mouse² and test huddling afterward. The latter experiments are presently possible only in mice, and we used rats for our study. We are currently setting up huddling experiments in mice, but the study of huddling as we performed in our manuscript is not readily translatable from rats to mice. Moreover, utilizing genetically modified lines to knock down each of the 4 5HT1 receptor subtypes expressed in the thalamus, would have required a great number of animals from different strains, which goes beyond what our animal legislation, animal facility allowance, and financial support currently allow us.

- 2) *Linking neural circuits to behavior is not trivial, and so the authors are to be commended for their efforts. That being said, the authors exhibit a rather unnecessary tendency to overstretch their interpretations. The entire discussion around the idea of the 'steady, ready, go' organization of the cortex (pages 11 and 12) is purely speculative and is not based on experimental evidence.*

To address the reviewers' concern, we have now made clear that what we are discussing in the manuscript is our own *speculations* based on the current literature and our own data. Moreover, we decided not to move Supplementary figure 8 from the supplemental material to the main text as suggested by referee 1, not to strengthen further the reviewers' 3 concerns on our the 'steady, ready, go' model.

- 3) *There is not a single piece of evidence in the paper suggesting that the somatosensory cortex is required for feeding during early postnatal development, for example.*

To address the reviewers' concern, we have now made clear that what we are discussing in the manuscript is our own *speculations* based on the current literature.

- 4) *The suggestion that projections from the somatosensory cortex to limbic regions of the cortex mediate the function of S1 in huddling is also speculative and could have been tested experimentally.*

To address the reviewers' concern, we have now made clear that what we are discussing in the manuscript is our own *hypothesis* based on the current literature. As for point number 1 above, testing experimentally that projections from the somatosensory cortex to limbic regions of the cortex mediate the function of S1 in huddling would be presently possible only in mice, and we used rats for our study. We are currently setting up huddling experiments in mice, but the study of huddling as we performed in the manuscript is not ready translatable from rats to mice.

Bibliography

1. Bonnin, A., Torii, M., Wang, L., Rakic, P. & Levitt, P. Serotonin modulates the response of embryonic thalamocortical axons to netrin-1. *Nat Neurosci* **10**, 588–597 (2007).
2. Chen, X. *et al.* Disruption of Transient Serotonin Accumulation by Non-Serotonin-Producing Neurons Impairs Cortical Map Development. *Cell Rep.* **10**, 346–358 (2015).
3. Paolino, A., Fenlon, L. R., Suárez, R. & Richards, L. J. Transcriptional control of long-range cortical projections. *Curr Opin Neurobiol* **53**, 57–65 (2018).